# Effects of Groundwater Depth on Vegetation Coverage in the Ulan Buh Desert in a Recent 20-Year Period

Ting Lu [1,2,3], Jing Wu [1,3,4], Yangchun Lu [1,2,3], Weibo Zhou [1,3] and Yudong Lu [1,2,3,*]

1    Key Laboratory of Subsurface Hydrology and Ecological Effects in Arid Region of the Ministry of Education, Chang'an University, Xi'an 710054, China; 2019029004@chd.edu.cn (T.L.); endless5wj@163.com (J.W.); 18142399823@163.com (Y.L.); zwbzyz823@163.com (W.Z.)
2    Key Laboratory of Mine Geological Hazards Mechanism and Control, Ministry of Natural Resources, Xi'an 710054, China
3    School of Water and Environment, Chang'an University, Xi'an 710054, China
4    Qinghai Institute of Geo-Environment Monitoring, Xining 810008, China
*    Correspondence: luyudong@chd.edu.cn

**Abstract:** As a typical desert in the Inner Mongolia Autonomous Region, the Ulan Buh Desert has a dry climate and scarce precipitation all year round. Groundwater has become the main factor limiting the growth of vegetation in this region. It is of great significance to study the influence of groundwater depth on the spatial distribution pattern of vegetation in this region. Based on the PIE-Engine platform and using long-term time-series Landsat data, this paper analyzed the spatial–temporal distribution characteristics and trends in vegetation coverage in the Ulan Buh Desert in the last 20 years using a pixel dichotomy model and the image difference method. The Kriging interpolation method was used to interpolate the groundwater depth data from 106 monitoring wells in the Ulan Buh Desert over the past 20 years, and the spatial distribution characteristics of groundwater depth in the Ulan Buh Desert were analyzed. Finally, the correlation coefficient between changes in vegetation coverage and changes in groundwater depth was calculated. The results showed the following: (1) The vegetation coverage in the Ulan Buh Desert was higher in the periphery and lower in the center of the desert. The overall vegetation level showed an increasing trend year by year; the growth rate was 4.73%/10 years, and the overall vegetation cover showed an improving trend. (2) The overall groundwater depth in the Ulan Buh Desert was deep in the southwest and shallow in the northeast. In the past 20 years, the groundwater depth in the Ulan Buh area has become shallower, and the ecological condition has gradually improved. (3) On the whole, the vegetation coverage varied with the groundwater depth, and the shallower the groundwater depth, the greater the vegetation coverage. When the groundwater depth increased to more than 4 m, the change in the groundwater depth had a significant effect on the vegetation coverage. However, when the groundwater depth was greater than 6 m, the change in the groundwater depth had no significant effect on the change in vegetation coverage.

**Keywords:** Ulan Buh Desert; vegetation coverage; pixel binary model; groundwater depth; correlation coefficient

## 1. Introduction

Vegetation has always been the material basis for human survival and development [1]. It is also an important part of the terrestrial ecosystem and plays an irreplaceable role in the sustainable development of global and regional ecosystems [2–4]. In recent years, climate change [5,6] and human activities [7–9] have significantly altered the dynamics of terrestrial plants.

There has been a large amount of research on surface water in various regions of the world [10–13], but relatively little research on groundwater. Inner Mongolia is in a transition zone from a humid area to an arid and semi-arid area in the north of China, with

an uneven distribution of water resources and great variations in runoff [14,15], both of which are very sensitive to changes in the ecological environment, making it one of the ideal regions to study changes in regional vegetation [16,17]. As a typical desert in Inner Mongolia, the Ulan Buh Desert has attracted the attention of many scholars for a long time. Due to its proximity to the border of the East Asian summer monsoon region, this region is more sensitive to the fluctuation in monsoon intensity and is one of the most seriously desertified areas in China. The climate in this region is arid all year long with minimal precipitation [18,19], and the growth and development of vegetation are highly dependent on the groundwater burial depth [20–22]. Exploring the relationship between vegetation and groundwater burial depth in this region can provide a certain reference value for the study of desert vegetations in northern China.

Groundwater has potential ecological consequences in the Ulan Buh Desert [23–25]. When the groundwater depth is deep, the soil moisture content becomes low, and vegetation growth is limited. When the groundwater depth is shallow, the soil moisture content increases, and the vegetation biomass changes dramatically under the action of groundwater capillarity. However, when the burial depth is too shallow, the salinization of shallow soil will inhibit plant growth to a certain extent [26,27]. Therefore, it is of great significance for vegetation protection in the Ulan Buh area to conduct the quantitative analysis of vegetation coverage in relation to different buried depths and explore the influence of groundwater depth on the vegetation spatial distribution pattern.

At present, there are two main methods used to study the correlation between groundwater depth and vegetation. One is to study the appropriate ecological water level of different vegetation populations after vegetation population division in the study area [28,29]. Cheng Yan et al. [30] adopted a vegetation quadrat survey to obtain vegetation feature information in the study area and used a Gaussian model to conduct the statistical analysis of vegetation features and groundwater depth. Zhang et al. [31] statistically analyzed the critical water level of ecological vegetation succession after groundwater development using the vegetation structure map analysis method. Although these methods are simple, intuitive, and highly applicable, field investigations require a lot of manpower and material resources, and the accuracy of the samples directly determines the reliability of the results [32]. The second method is to use the vegetation index as a regional ecological evaluation factor to study the response relationship between it and the groundwater depth [33–35]. Jin et al. [34] analyzed the correlation between the *NDVI* (normalized difference vegetation index) and groundwater depth in the Yinchuan Plain and found that the groundwater level depth had a significant control effect on vegetation growth. Song et al. [36] explored the correlation between vegetation and various influencing factors in a desert grassland area in Inner Mongolia using multiple sources of remote sensing satellite data and groundwater data. Within a certain threshold range, there is a clear linear relationship between the *NDVI* and groundwater depth. As a measure of the surface vegetation cover condition [37,38], vegetation coverage can reveal things about the regional ecological environment and evaluate the regional ecological quality [39,40]. At present, a large number of studies have shown that the *NDVI* is the most commonly used variable and is a highly useful index to calculate vegetation coverage [41–43].

This study used long-term Landsat data to extract vegetation coverage information using the PIE (Pixel Information Expert) Engine platform in order to study the changes in the vegetation cover in the Ulan Buh Desert over the past twenty years. The Kriging interpolation method was used to process the measured groundwater data in the Ulan Buh Desert, and the correlation coefficient between vegetation coverage and the groundwater depth was calculated to clarify the spatial response relationship between vegetation coverage and groundwater depth in the Ulan Buh area in order to provide a scientific basis for vegetation restoration and groundwater resource management in the Ulan Buh area.

## 2. Materials and Methods

### 2.1. Study Area

The Ulan Buh Desert (Figure 1) is one of the major deserts in China. It is part of the northwest desert region of our country, located in the Alashan League and Bayannur City in the west of the Inner Mongolia Autonomous Region, bordering the Yellow River in the east, reaching the northern foothills of the Helan Mountain in the south, and extending to the Wolfshan-Bayannur Mountain Range in the west [44]. The elevation of the Ulan Buh Desert area ranges from 971 to 1353 m, with a relative elevation difference of 340 m, and the relative height of local dunes can reach 50–60 m [18]. The landform is mainly dominated by denudation hills, accumulated platforms, accumulated basins, the Yellow River valley, and alluvial plains. The Ulan Buh Desert is located in the temperate, semi-arid to arid climate transition zone. The climate is characterized by sufficient sunshine, little rain, hot summers, cold winters, large daily temperature differences, strong evaporation, strong winds, and a short frost-free period. The average annual temperature reaches 8.6 °C. The highest temperature in July is 38.7 °C, the lowest temperature in January is −32.8 °C, the average annual precipitation reaches 116–162 mm, and the average annual evaporation is 2560–3200 mm [45,46]. The main vegetation is sea buckthorn, sand holly, white thorn, overlord, red yarn, and reed.

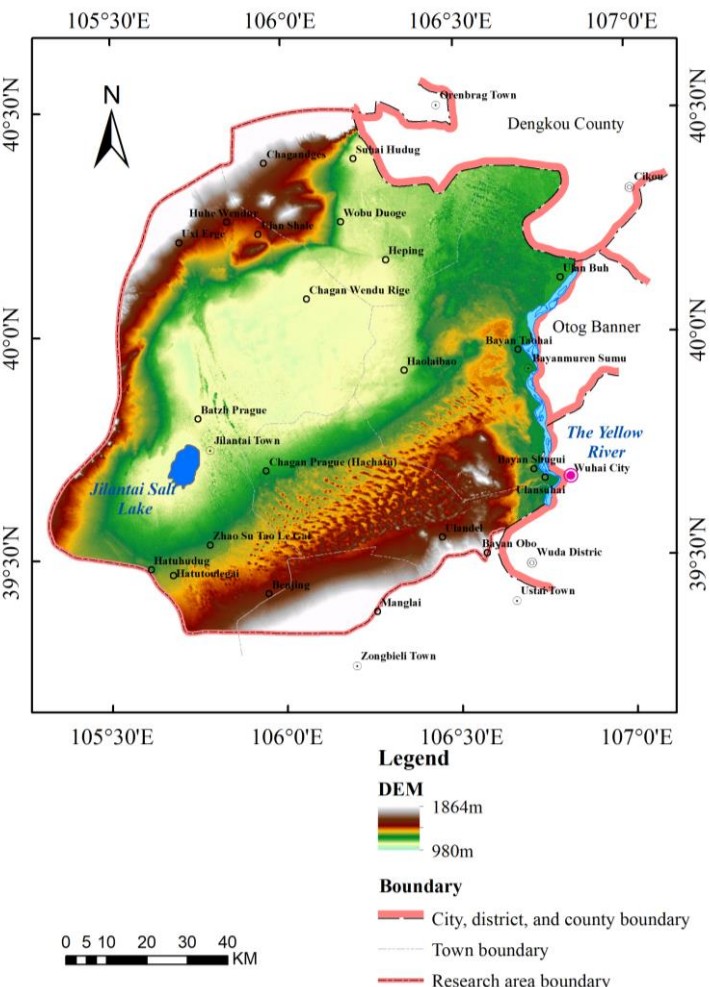

**Figure 1.** Overview of the study area.

### 2.2. Data Source and Processing

The Ladsat-5 multispectral data and Ladsat-8 multispectral data used in this paper are from the Landsat program of the National Aeronautics and Space Administration (NASA), SRTM (Shuttle Radar Topography Mission, which was established with support from the

German Aerospace Center (DLR)), and the Italian Space Agency (ASI). An international project was formed by NASA and the National Geospatial-Intelligence Agency (NGA) to acquire ordnance survey data through space shuttle-mounted radar [47]. The meteorological data are from the National Meteorological Information Center (http://data.cma.cn/ accessed on 3 April 2023), using average annual temperatures and precipitation from 2000 to 2020.

The groundwater monitoring data used in this study consist of the measured groundwater depth levels at 107 monitoring wells in Jilantai, Chaganwendugrige, Maiwulajia, Bayanaobao, Heping, and other regions in 2000, 2005, 2010, 2015, and 2020 [48].

The PIE-Engine platform is a professional PaaS/SaaS cloud computing service platform built on container cloud technology and independently developed by Aerospace Hongtu, which is similar to the GEE (Google Earth Engine) and has powerful data storage and high-performance analysis and calculation capabilities. Cheng Wei demonstrated the effectiveness of the spatiotemporal remote sensing cloud computing platform of the PIE-Engine Studio in their paper "Research and Application" [49]. Remote sensing data used in this study were processed online with the PIE-Engine platform, including Landsat-5 multispectral data, Landsat-8 multispectral data, and SRTM digital elevation data. The optical composite images were created from data obtained during the months of August and September in 2000, 2005, 2010, 2015, and 2020. This period was chosen because it provides the most cloud-free data and is within the growth period of vegetation, which can retain more vegetation information. To mitigate the effects of cloud pollution, the percentage of cloud cover was limited (<20%) when synthesizing cloudless images. Then, the Landsat cloud mask algorithm was used to calculate the image in the specified time and space range, and the median synthesis method was used to reconstruct the minimum cloud coverage composite image. Benefiting from the PIE-Engine platform's data operation and management mechanism, all remote sensing data used in this study were sampled to 30 m, and the PIE-Engine ensured geometric registration accuracy between different data sources by using a unified coordinate system based on the embedded algorithm. Using the PIE-Engine platform, the near-infrared band (NIR, 0.76~0.96 μm) and visible *RED* band (*RED*, 0.62~0.69 μm) were calculated to obtain the *NDVI* of each pixel in the study area. Finally, the *NDVI* remote sensing images with outliers removed were obtained with the calculation formula of outliers.

*2.3. Research Method*

2.3.1. Calculation of *NDVI*

The *NDVI* is used as a long-term monitoring tool to evaluate the growth status of plant coverage and is the most common standardized method to measure vegetation cover [50]. It is sensitive to vegetation growth and change and is the most commonly used index by analysts at present [51]. Formula (1) for calculating the *NDVI* is as follows:

$$NDVI = \frac{NIR - RED}{NIR + RED} \tag{1}$$

*NIR* is the near-infrared wave band, and *RED* refers to infrared wave band. This research adopts the Landsat TM/ETM images' *RED*-corresponding wave band. The Landsat OLI images' *NIR*-corresponding band is Band5 and the *RED*-corresponding band is Band4.

For outliers, with *NDVI* values greater than 1 and less than −1, a mask calculation on the PIE-Engine platform was used to remove them.

2.3.2. Estimation of Vegetation Coverage

The pixel binary model is a simple and practical remote sensing estimation model. It is often used to calculate vegetation cover because it can reduce the influence of atmosphere and water on remote sensing images [52]. The principle of the binary pixel model is to

divide the spectral information of a remote sensing image into two parts, namely vegetation cover and no vegetation cover. The specific calculation formula is as follows:

$$VFC = \frac{S - S_{soil}}{S_{veg} - S\_soil} \tag{2}$$

Here, $VFC$ is vegetation coverage, $S$ is mixed pixels information, $S_{soil}$ means no vegetation information, and $S_{veg}$ indicates vegetation-like meta information. When the binary pixel model is used to analyze vegetation information, the $NDVI$ is usually used as an estimate. Replacing pixel information with the $NDVI$ can reduce the error caused by radiation. The formula is as follows:

$$VFC = \frac{NDVI - NDVI_{soil}}{NDVI_{veg} - NDVI_{soil}} \tag{3}$$

$NDVI_{soil}$ means no vegetation information and $NDVI_{veg}$ indicates vegetation-like meta information. Usually, many scholars intercept the upper and lower thresholds of the $NDVI$ within a certain confidence interval according to the gray distribution of the $NDVI$ in the whole image to approximately represent $NDVIv$ and $NDVIs$ [53,54]. In this paper, according to the frequency statistical chart of $NDVI$ data, the $NDVI$ value with a cumulative frequency of 5% is taken as $NDVIs$, and the $NDVI$ value with a cumulative frequency of 95% is taken as $NDVIv$.

Five periods of remote sensing images from 2000 to 2020 were processed based on the PIE-Engine platform, and five sets of vegetation coverage data were obtained in 2000, 2005, 2010, 2015, and 2020. The vegetation coverage data were processed and the change trend of the vegetation coverage in the Ulan Buh Desert was analyzed.

According to the vegetation characteristics of the Ulan Buh Desert and the technical regulations of the Land Use Status Investigation, Technical Specifications of Chinese Desert Cataloging and National Ecological and Environmental Standards of the People's Republic of China [55], the vegetation coverage of the Ulan Buh Desert was divided into four levels: very low coverage ($0 \leq VFC < 0.2$), low coverage ($0.2 \leq VFC < 0.3$), medium coverage ($0.3 \leq VFC < 0.6$), and high coverage ($0.6 \leq VFC < 1$).

### 2.3.3. Difference Comparative Analysis

The image difference method involves subtracting or dividing the remote sensing images of two time phases. The principle is that the unchanged part of the image generally has an equal or similar gray value in the remote sensing image of the two phases, and when the two images change, the gray value of the corresponding position will be greatly different. It can be conducted using grayscale values or feature values to obtain the difference image [56]. The spatiotemporal changes in vegetation in the study area could be obtained through the image difference method, and the positive and negative values of the difference could reflect the increase or decrease in vegetation [57]. A positive difference indicates an increase in vegetation in the study area, a negative difference indicates a decrease in vegetation in the study area, and a difference of 0 indicates no change in vegetation cover status. The specific formula is as follows (4):

$$\Delta VFC = VFC_{year2} - VFC_{year1} \tag{4}$$

In the formula, $\Delta VFC$ represents the change in vegetation coverage, $VFC_{year2}$ represents the vegetation coverage in the following year, and $VFC_{year1}$ represents the vegetation coverage in the previous year. By comparing the vegetation cover map of the Ulan Buh Desert in 2000 with the vegetation cover map in 2020, the spatial changes in the vegetation cover of the Ulan Buh Desert were obtained. According to the method of standard deviation, the spatial change in vegetation cover in the Ulan Buh Desert was divided into 7 levels, including extreme improvement ($0.67 \leq \Delta VFC < 1$), moderate improvement ($0.33 \leq \Delta VFC < 0.67$), slight improvement ($0 \leq \Delta VFC < 0.33$), unchanged ($\Delta\Delta VFC = 0$),

slight decline ($-0.33 \leq \Delta VFC < 0$), moderate decline ($-0.67 \leq \Delta VFC < -0.33$), and extreme decline ($-1 \leq \Delta VFC < -0.67$).

### 2.3.4. Data Gridding

Data gridding refers to the method of converting point positioning data into surface data through spatial topology analysis [58]. The purpose of gridding is to make each data point more standardized for statistical purposes. This article used the fishing net tool and a square grid as the grid shape for the data grid. A total of 1703 regular square grids with an area of 6.25 km$^2$ were established within the study area.

### 2.3.5. Kriging Interpolation

Kriging interpolation is based on the concept of spatial autocorrelation, that is, the closer the points are, the stronger the correlation between them. It is a method of assigning weight to each sample according to its spatial distribution position and the degree of correlation between the samples, and it estimates the average value of the samples on unknown sample points in a weighted-average manner [59]. The ordinary Kriging interpolation method is the most basic and widely used interpolation method among all Kriging interpolation methods. It first considers the variation distribution of spatial attributes in the spatial position, determines the distance range that affects the value of a point to be interpolated, and then estimates the attribute value of the point to be interpolated with the sampling points in this range. Its basic principle is to estimate data from other unobserved positions in space through regularly distributed sample data. Therefore, it is necessary to fit an empirical semi-variogram model to reflect the relevant characteristics of the spatial data, and then obtain weights for prediction [60]. The calculation formula for the most basic semi variogram is as follows (5) [61]:

$$\gamma(h) = \frac{1}{2N(h)} \sum_{i=1}^{N(h)} [z(x_i + h) - z(x_i)]^2 \tag{5}$$

In the formula, $h$ is the sample spacing, $N(h)$ is the logarithm of sample points separated by a distance $h$ in space, and $z(x_i)$ and $z(x_i + h)$ are the variable values at points $x_i$ and $x_i$+$h$, respectively. For $\gamma(h)$, as the sample spacing $h$ increases, the half square difference of all lag distance pairs reaches a relatively stable constant value from a non-zero value.

This study used Kriging interpolation to analyze the spatial distribution characteristics of groundwater depth data from 106 monitoring wells in the Ulan Buh Desert region from 2000 to 2020 and summarized the spatiotemporal changes in groundwater depth in the region over a five-year period. This study aimed to analyze the correlation between groundwater depth and vegetation coverage.

### 2.3.6. Correlation between Groundwater Depth and Vegetation Coverage

Correlation analysis refers to the analysis of two or more correlated variable elements to measure the degree of correlation between the two factors [62]. In order to study the impact of groundwater depth on vegetation coverage, this study used pixels as the calculation unit to calculate the correlation coefficients of vegetation coverage changes and groundwater depth changes between 2000 and 2020. The calculation Formula (6) is as follows [63–66]:

$$R = \frac{\Sigma(x_i - \bar{x})(y_i - \bar{y})}{\Sigma(x_i - \bar{x})^2 \Sigma(y_i - \bar{y})^2} \tag{6}$$

In the formula, $x_i$ represents the vegetation coverage in 2000 and 2020, $y_i$ represents the groundwater depth in 2000 and 2020, $\bar{x}$ is the average vegetation coverage over the past 20 years, and $\bar{y}$ is the average groundwater depth over the past twenty years.

## 3. Results and Discussion

*3.1. Temporal and Spatial Variation in Vegetation Coverage*

3.1.1. Temporal Distribution Characteristics of the Vegetation Cover

The trend of vegetation cover change in the Ulan Buh Desert over the past 20 years is shown in Figure 2. During the period from 2000 to 2020, the annual average vegetation cover of the Ulan Buh Desert showed an overall increasing trend year by year, ranging from 0.30 to 0.46, with an increase rate of 4.73% per decade. The highest vegetation cover occurred in 2020, with a vegetation coverage of 0.4560. The lowest vegetation coverage occurred in 2000, with a coverage of 0.3377. The vegetation coverage showed a slight downward trend in 2010. Except for 2010, the vegetation coverage in other years showed an upward trend compared to the previous period, and the growth rate of vegetation coverage was the fastest from 2000 to 2020, with a growth rate of 11.88% per decade.

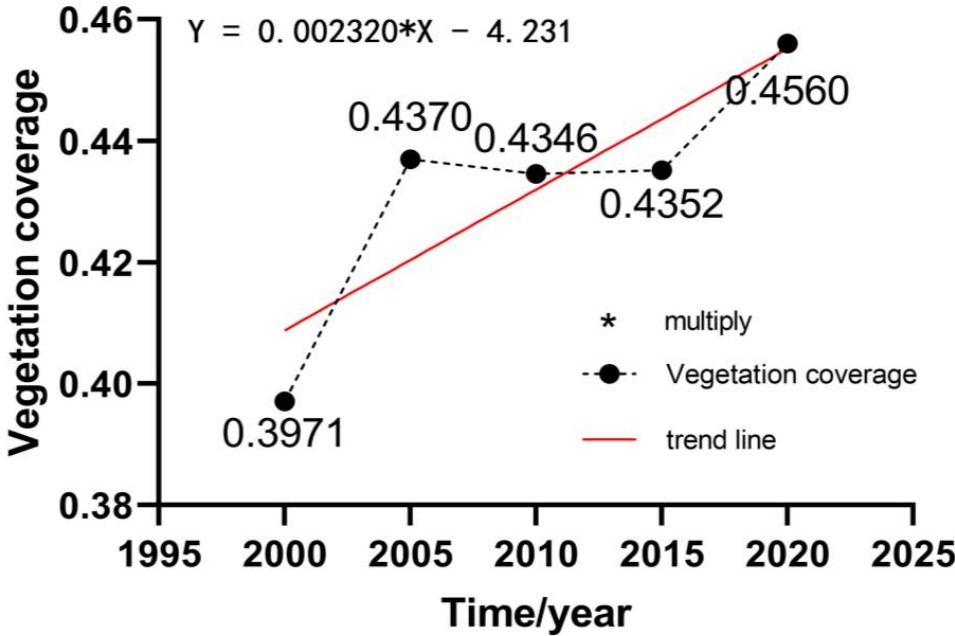

**Figure 2.** Interannual Changes in Vegetation Coverage.

3.1.2. Spatial Dynamic Change Characteristics of the Vegetation Cover

Through the analysis of the vegetation coverage classification map (Figure 3) of the Ulan Buh Desert in 2000, 2005, 2010, 2015, and 2020, it was found that the degree distribution of vegetation coverage in each year in the study area showed a certain regularity, that is, the vegetation coverage of the Ulan Buh Desert was generally distributed in a large area, with high coverage around the outer sections and low coverage in the middle, and the high vegetation coverage was mainly concentrated in Aolunbulag Town, Jilantai Salt Lake, and the surrounding areas of the Yellow River basin. The vegetation coverage in the central region was low, especially in the towns of Qulantai and Ustai, where the middle and low vegetation coverage areas surrounded the growth of low vegetation coverage areas. The vegetation coverage level gradually increased as it diverged outward from the low vegetation coverage area.

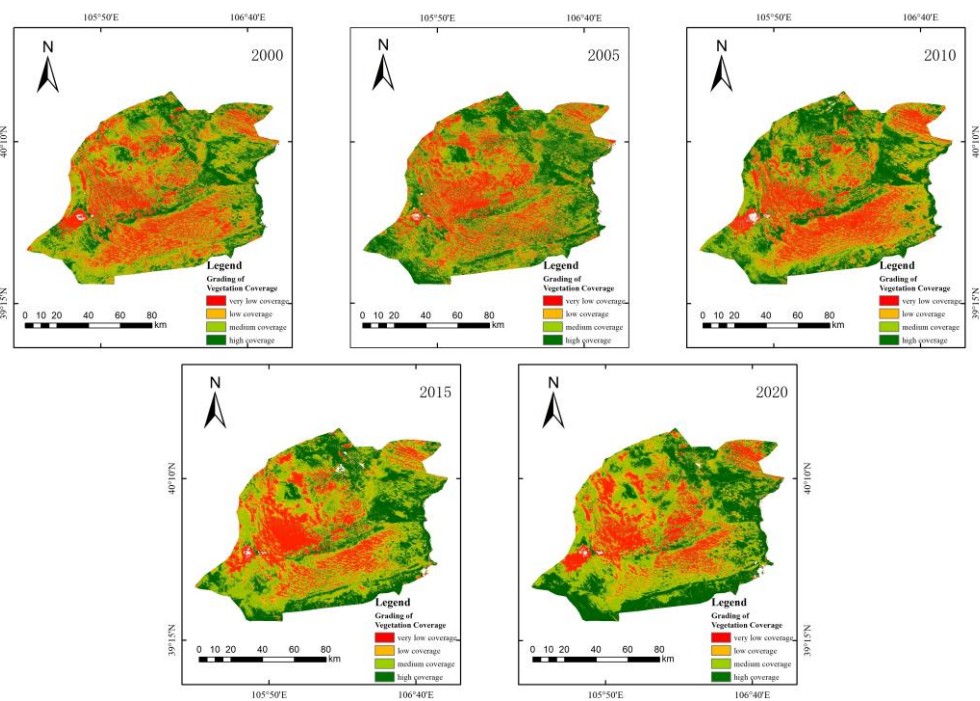

**Figure 3.** Vegetation Coverage Grading Maps of the Ulan Buh Desert.

### 3.1.3. Analysis of the Vegetation Coverage Change Trend

The absolute spatial distribution map of vegetation coverage in the Ulan Buh Desert is shown in Figure 4. The overall vegetation coverage in the Ulan Buh Desert showed an improvement trend, with 67.76% of the areas showing improvement and 2.47% of the areas showing extreme improvement. It was mainly distributed in patches near the Yellow River basin and many urban residential areas. The proportion of areas with moderate improvement was 16.79%, mainly distributed around extreme improvement areas, with distribution at both ends and boundaries. The areas with slight improvement accounted for 48.50%, spread over the entire Ulan Buh Desert, most widely distributed in the central and southern regions. The proportion of areas with declining vegetation coverage was 32.23%, with areas of extreme decline accounting for 0.29%. They were mainly distributed at the upper boundary of the Ulan Buh Desert and presented a local patchy pattern. The proportion of areas with moderate recession was 4.03%, mainly distributed in the upper boundary area; however, there was also a scattered distribution in the lower area. The proportion of areas with a slight recession was 27.91%, distributed above the central region, presenting a large-scale blocky distribution. It can be seen that areas showing improvement were greater than those showing a decline, and the overall vegetation coverage of the Ulan Buh Desert showed an improvement trend.

### 3.2. Changes in Groundwater Depth

In this study, five time nodes (2000, 2005, 2010, 2015, and 2020) were selected to obtain groundwater depth data from groundwater level monitoring wells in the Ulan Buh Desert, and spatial interpolation was carried out to obtain the spatial variation characteristic maps of the groundwater level in the Ulan Buh Desert from 2000 to 2020, as shown in Figure 5.

From a spatial perspective, the groundwater depth in the Ulan Buh Desert was deep in the southwest and shallow in the northeast. The underground water depths in the central and northeast regions of the Ulan Buh area were shallow, with depths of 2–4 m in most regions and 0–2 m in some regions. The depth of groundwater in the southwest was relatively deep; the depth of groundwater in most areas was more than 6 m, and the depth in the farthest southern region was more than 10 m.

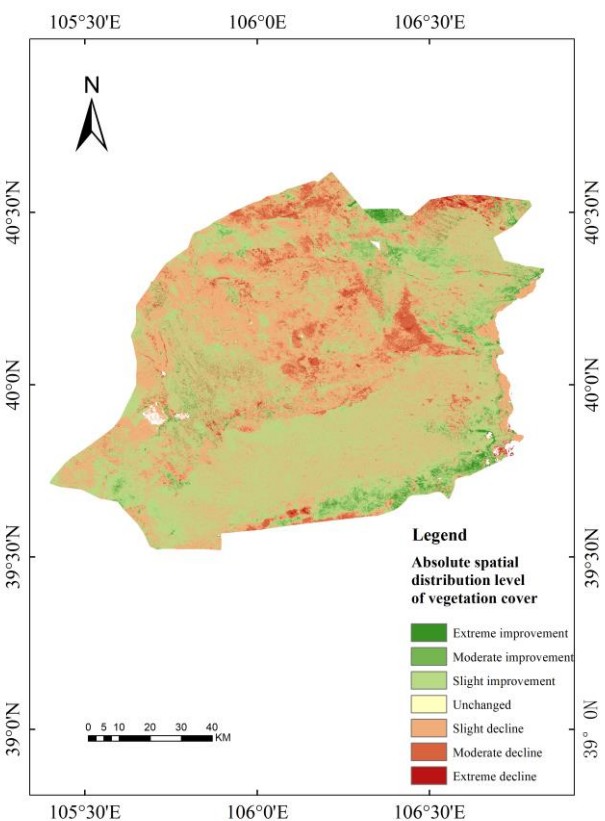

**Figure 4.** Absolute spatial distribution of vegetation coverage in the Ulan Buh Desert.

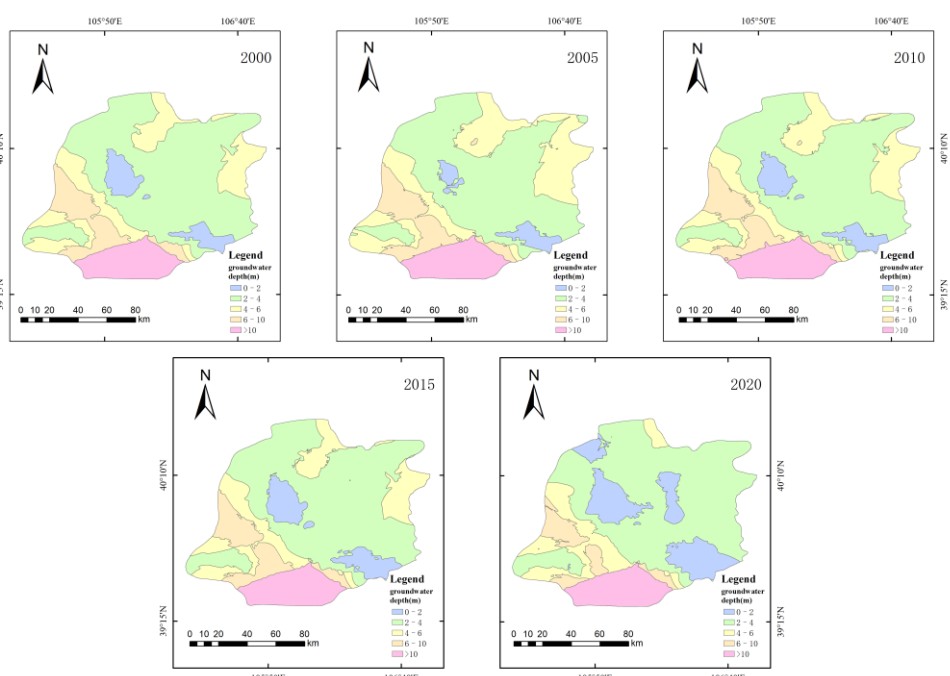

**Figure 5.** Groundwater Depth from 2000 to 2020.

From the perspective of time, from 2000 to 2020, the area with groundwater depth less than 4 m increased by 1280 m$^2$, accounting for about 11% of the total area of the Ulan Buh Desert, which indicates that the overall groundwater depth of the Ulan Buh Desert showed a trend of becoming shallower. However, from 2000 to 2005, the groundwater depth became deeper. The area of 0–2 m groundwater depths decreased significantly, from

726 m$^2$ in 2000 to 464 m$^2$ in 2005, a reduction of about 36%, and the area of 2–4 m buried depths also decreased by 674 m$^2$. The areas with underground water depths of 4–6 m and 6–10 m increased to a certain extent, and the underground water depth of the Ulan Buh Desert showed a trend of deepening from 2000 to 2005. This may be due to the continuous decrease in average precipitation between 2001 and 2003, as shown in Figure 5.

From the changes in groundwater depth from 2000 to 2020, we can see that, except for the years 2000–2005, the groundwater depth of the Ulan Buh Desert showed an overall trend of becoming shallower. Considering the changes in vegetation coverage in the above section, it is not difficult to see that the ecological situation of the Ulan Buh Desert showed a gradual trend of improvement. Moreover, the shallower the buried area, the more sensitive it was to the change in climate and precipitation, being more likely to change under the influence of climate and precipitation and other factors.

### 3.3. Correlation between Changes in Groundwater Depth and Vegetation Coverage

In order to study the relationship between the groundwater depth change and vegetation coverage change, this paper first obtained the difference in groundwater depth and vegetation coverage of each pixel in the two time nodes of the study area in 2020 and 2000 through difference calculation, then normalized the two difference values to calculate the correlation coefficient between them. The distribution of correlation coefficients is shown in Figure 6. It can be seen that the proportion of areas with positive correlation between vegetation coverage changes and groundwater depth changes is 53%, while the proportion of areas with negative correlation is 47%. It can be found that when the groundwater depth rose to more than 4 m, the change in groundwater depth had a significant effect on the vegetation coverage. When the groundwater depth was greater than 6 m, the change in groundwater depth had no obvious effect on the change in vegetation coverage.

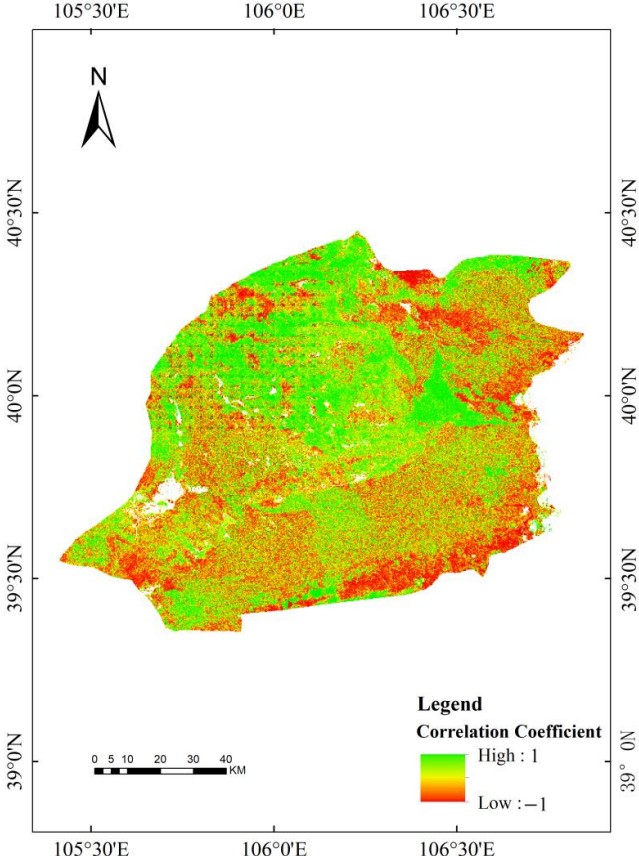

**Figure 6.** Distribution of correlation coefficients between changes in groundwater depth and vegetation coverage in the Ulan Buh Desert.

Using 2020 groundwater depth data and 2020 vegetation coverage data, taking vegetation coverage as the horizontal coordinate and groundwater depth data obtained from monitoring wells as the vertical coordinate, a rectangular coordinate system was established to obtain the scatter plot in Figure 7. This scatter plot reflects the influence of groundwater depth on the vegetation index. It can be seen that under the same groundwater depth, there were great differences in vegetation coverage, which can be caused by climate, soil type, vegetation type, and other factors. However, overall, there was a certain regularity in the variation in vegetation coverage with the depth of groundwater. The shallower the groundwater depth, the greater the vegetation coverage. Through analysis, it can be found that when the groundwater depth increased to over 4 m, the change in groundwater depth had a significant improvement effect on vegetation coverage. When the groundwater depth was greater than 6 m, there was no significant impact of changes in groundwater depth on vegetation coverage.

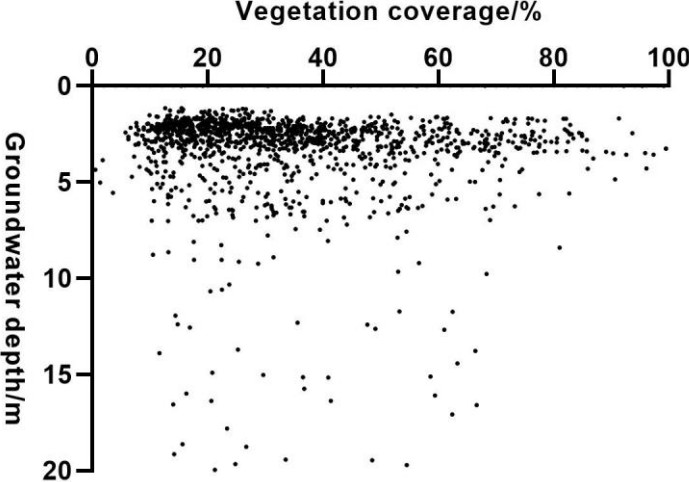

**Figure 7.** Characteristics of vegetation coverage changes with groundwater in the Ulan Buh Desert research area.

*3.4. Discussion*

3.4.1. Influence of Temperature

Li Yan [67] explored the relationship between soil temperature and soil moisture content in the Ulan Buh area and utilized an evaluation index system for soil moisture content, groundwater depth, capillary rise height of sandy soil, annual variation in groundwater depth, groundwater mineralization, and soil salt content. The ecological suitability of vegetation was analyzed, and the suitable area was divided into more suitable areas, less suitable areas, and unsuitable areas. Several studies have proven that the growth suitability of vegetation is related to temperature and groundwater depth. Reza Amiri [68] and others retrieved the surface temperature of the Tabriz metropolitan area in Iran using Landsat satellite data and established the temperature vegetation index (TVX). The results showed that, over time, the TVX had migrated from low temperature dense vegetation to high temperature sparse vegetation, indicating that the heat island effect generated by urban development had to some extent affected the growth of vegetation. Based on the above article, we investigated the response of the vegetation in the Ulan Buh area to temperature and groundwater.

As shown in Figure 8, From 2000 to 2017, the average annual temperature in the Ulan Buh area showed an overall upward trend, with some fluctuations over the years. Due to the cold resistance, heat resistance, dryness preference, and strong ecological adaptability of vegetation in desert areas, a slight increase in temperature can promote the growth of vegetation in desert areas to some extent.

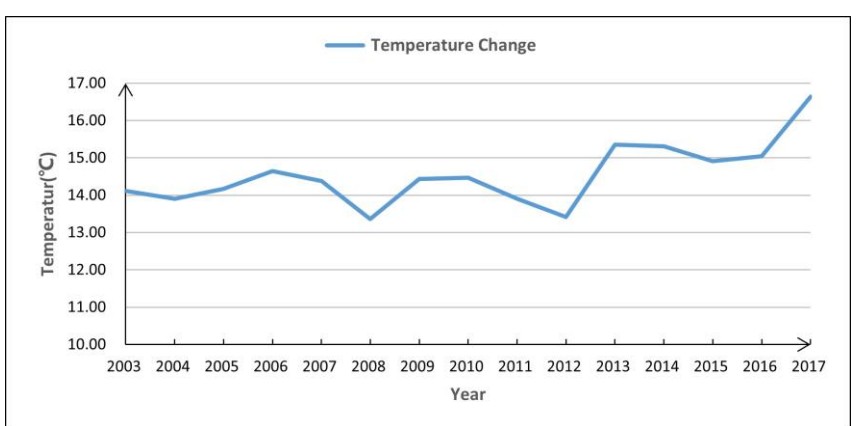

**Figure 8.** Trend of temperature changes in the Ulan Buh Desert.

### 3.4.2. Influence of Precipitation

Figure 9 shows a line chart of average precipitation for the years 2000–2020. It can be seen that the precipitation decreased significantly from 2001 to 2003. At the same time, compared with 2000, the vegetation coverage decreased in 2005, and the groundwater depth also showed the same trend. The area with the most suitable groundwater depth decreased by 674.3 km². From 2005 to 2008, the precipitation was at a relatively high level, and the vegetation growth trend was good. The area with the most suitable groundwater burial depth increased by 498.8 km². After 2011, the annual precipitation stabilized, vegetation coverage also steadily increased, and the most suitable areas for groundwater burial slowly increased. It is not difficult to see that an appropriate increase in precipitation contributes to the growth of vegetation. It is reasonable to infer that precipitation affects the depth of groundwater and thus affects the growth of vegetation.

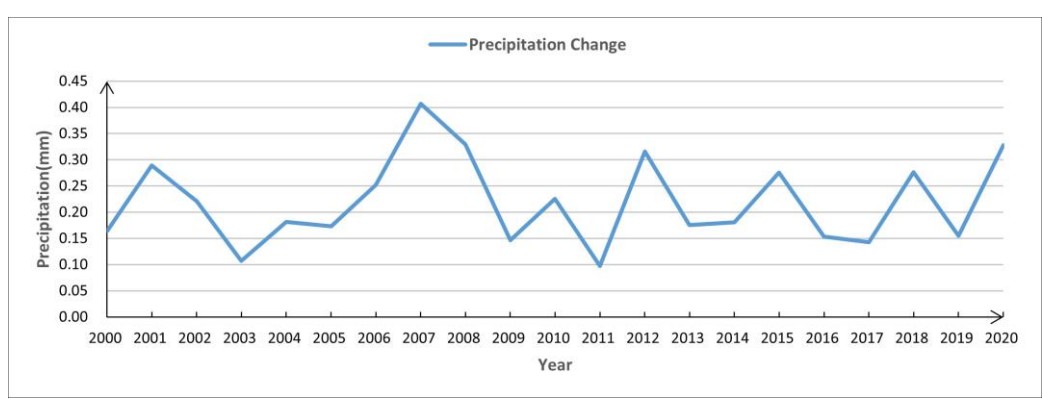

**Figure 9.** Trend of precipitation changes in the Ulan Buh Desert.

### 3.4.3. Other Influencing Factors and Analysis

The eastern and northern parts of the Ulan Buh Desert are close to the Yellow River; therefore, the local government has been carrying out wind prevention, sand fixation, and afforestation projects around the edge of the Ulan Buh Desert for many years. This has resulted in high vegetation coverage in the periphery of the Ulan Buh Desert and low vegetation coverage in the center of the desert.

In this paper, the Ulan Buh Desert, a typical desert in northern China, was selected as a research area. By revealing the correlation between the groundwater depth change and vegetation coverage change in the Ulan Buh Desert, the influence mechanism of groundwater depth on vegetation in desert areas can be obtained. Through vegetation as a key ecological environment element, we can also obtain the general change trend of the

local ecological environment, which can provide a certain reference for the research and analysis of the ecological environment in similar desert areas.

### 3.4.4. Limitations

There are some limitations in the Landsat data used in this paper. Future studies can use data with higher accuracy, such as Sentinel data, to reduce the errors that may be introduced in the process of data acquisition and processing. When using the Kriging interpolation method, it is necessary to consider whether the attribute is stationary in space and whether the sample points are evenly distributed to deal with the uncertainty. In order to increase the reliability and accuracy of the research results, subsequent research can use multi-source data combined with different interpolation methods. Future studies can add more influencing factors for analysis, such as human factors, soil factors, etc., to further reveal the influencing mechanism behind the change in the desert ecological environment.

### 4. Conclusions

This study used the PIE-Engine software for spatial interpretation, quantitatively analyzed the spatiotemporal changes in vegetation coverage and water depth in the Ulan Buh Desert over a 20-year period, and explored the mechanism of the response of groundwater depth to vegetation coverage.

(1)  In terms of time, the vegetation coverage of the Ulan Buh Desert has shown an overall trend of increasing year by year over the past 20 years, with an increase rate of 4.73%/10 years. The highest vegetation coverage appeared in 2020 and represented a 35% increase compared to the year with the lowest vegetation coverage (2000). The vegetation coverage in 2010 showed a slight downward trend. Except for 2010, the vegetation coverage in other years showed an upward trend compared to the previous period, and the growth rate of vegetation coverage was the fastest from 2000 to 2020, with a growth rate of 12% per decade. The downward trend of vegetation coverage in 2010 may have been influenced by precipitation and temperature.

(2)  In space, the degree distribution of vegetation coverage in the Ulan Buh Desert in each year showed a certain regularity. The vegetation coverage in the drainage basin was high around the periphery and low across a large area in the middle. The vegetation coverage of the Ulan Buh Desert showed an overall improvement trend, with 68% of the areas showing improvement in vegetation coverage, including 3% showing extreme improvement, 16.79% showing moderate improvement, and 48.50% showing slight improvement. The proportion of areas with a declining vegetation cover was 32%, with areas of extreme decline accounting for less than 1%, areas of moderate decline accounting for 4%, and areas of slight decline accounting for 28%. The overall vegetation coverage in the Ulan Buh Desert was relatively stable. The vegetation coverage in most areas of the Ulan Buh Desert basin showed a moderate-to-strong variation, with only a small number of areas experiencing a weak variation. Strong-variation areas were mainly distributed in the central region, while moderate variation areas almost covered the Yellow River, which flows through the Ulan Buh Desert region and the Salt Lake region. Weak variation areas were mainly scattered at the eastern and southwestern boundaries.

Based on the analysis of groundwater depth, there was a certain regularity in the variation in vegetation coverage with groundwater depth. The shallower the groundwater depth, the greater the vegetation coverage. The proportion of areas with a positive correlation between vegetation coverage changes and groundwater depth changes was 53%, while the proportion of areas with a negative correlation was 47%. When the groundwater depth increased to more than 4 m, the change in groundwater depth had a significant improvement effect on vegetation coverage. When the groundwater depth was greater than 6 m, there was no significant impact of changes in groundwater depth on vegetation coverage.

**Author Contributions:** Conceptualization, J.W. and T.L.; methodology, W.Z. and Y.L. (Yangchun Lu); investigation, T.L., J.W. and Y.L. (Yudong Lu); resources, W.Z.; data curation, J.W. and Y.L. (Yangchun Lu); writing—original draft preparation, T.L. and J.W.; writing—review and editing, Y.L. (Yudong Lu) and T.L.; project administration, Y.L. (Yudong Lu); funding acquisition, Y.L. (Yudong Lu). All authors have read and agreed to the published version of the manuscript.

**Funding:** This work was funded by the National Natural Science Foundation of China (Grant No. U2243204).

**Data Availability Statement:** Not applicable.

**Conflicts of Interest:** The authors declare no conflict of interest.

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
