# Peer review of "Effects of Groundwater Depth on Vegetation Coverage in the Ulan Buh Desert in a Recent 20-Year Period"

_water, doi:10.3390/w15163000_

Round 1
Reviewer 1 Report (Previous Reviewer 1)
The authors significantly improved the content of the manuscript.
Unfortunately, there is a problem with the literature, and it is unacceptable in its current state.
These are not major errors, but one gets the impression that the authors proceeded to improve the manuscript very chaotically.
1. A common mistake is to quote [2][3][4] when it should be [2-4]
lines: 65; 66; 70; 71; 75; 80; 86; 94; 102;103; 105; 136;200;
2. The text does not quote items from the list [7] [38], [44-46], [56].
3. The literature cited is not in the correct order.
Line 66 is the reference [10] and next line 70 is the reference [18]
Line 105 citing literature ends at [17] and then Line 113 is item [59] and where are the other items?
Please explain why in Section 3.2 Line 337 you refer to figure 9 from subsection 3.4
Author Response
Please see the attachment

Reviewer 2 Report (New Reviewer)
The paper titled "Influence of Groundwater Depth on the Spatial Distribution Pattern of Vegetation in Ulan Buh Desert" provides a comprehensive analysis of the relationship between groundwater depth and vegetation coverage in the Ulan Buh Desert. The study utilizes the PIE platform and long time series Landsat data to investigate the spatio-temporal distribution characteristics and trends of vegetation coverage over the past 20 years. The paper also includes an analysis of the spatial distribution of groundwater depth and examines the correlation between changes in vegetation coverage and groundwater depth.
Overall, the paper is well-structured and addresses an important research question regarding the impact of groundwater depth on vegetation in a desert region. The use of the PIE platform and Landsat data provides a robust foundation for the analysis. The findings of the study are presented clearly and concisely, and the conclusions drawn from the data are supported by the results.
One strength of the paper is the application of the pixel dichotomy model and image difference method to analyze the vegetation coverage. These methods help to identify the spatial distribution patterns and trends in vegetation over the study period. Additionally, the use of Kriging interpolation for groundwater depth data interpolation enhances the accuracy of the analysis.
The results indicate that vegetation coverage is higher in the periphery and lower in the center of the Ulan Buh Desert. The study finds an increasing trend in vegetation coverage over time, with a growth rate of 4.73% per 10 years. Moreover, the analysis shows a shallower groundwater depth in the northeast compared to the southwest, suggesting an improvement in the ecological condition over the past 20 years. The correlation between changes in vegetation coverage and groundwater depth demonstrates that shallower groundwater depths positively influence vegetation coverage, with a significant effect observed when the groundwater depth is less than 4m.
However, there are a few areas that could be further developed or clarified. Firstly, the paper would benefit from providing more details about the methodology employed in the pixel dichotomy model and image difference method. Additional information on how these methods were applied to the Landsat data would enhance the reproducibility of the study.
Furthermore, it would be helpful to discuss the limitations of the study and address potential sources of error or uncertainty in the analysis. For instance, the paper could mention any limitations in the accuracy or representativeness of the Landsat data, as well as the assumptions made in the Kriging interpolation method.
In conclusion, the paper makes a valuable contribution to the understanding of the relationship between groundwater depth.
Reference section should be increased with number of recent studies. I would like to suggest to author to include following published article to water related study of articles. May be cited in your work.
Groundwater Quality of Some Parts of Coastal Bhola District, Bangladesh: Exceptional Evidence
Toxicity and health risk assessment of polycyclic aromatic hydrocarbons in surface water, sediments and groundwater vulnerability in Damodar River Basin
Unregulated discharge of wastewater in the Mahanadi River Basin: risk evaluation due to occurrence of polycyclic aromatic hydrocarbon in surface water and sediments
Per-and polyfluoroalkyl substances in water and wastewater: A critical review of their global occurrence and distribution
Solvent Extraction Coupled with Gas Chromatography for the Analysis of Polycyclic Aromatic Hydrocarbons in Riverine Sediment and Surface Water of Subarnarekha River and Its…
little checking is required in introduction
Author Response
Please see the attachment

Reviewer 3 Report (New Reviewer)
The paper is well-written, and it is acceptable in its present form.
Please pay attention to the reference format (introduction section) and Figures 8 and 9; legends labels are missing.
No comments
Round 2
Reviewer 2 Report (New Reviewer)
Please find the enclosed reviewer comment in the attached file.

Author Response
Please see the attachment

This manuscript is a resubmission of an earlier submission. The following is a list of the peer review reports and author responses from that submission.
Round 1
Reviewer 1 Report
The article "Effects of groundwater depth on vegetation coverage in Ulan Buh Desert in recent 20 years" presented for review is interesting; the collection of data from the 20th anniversary seems impressive.
However, the preparation of the manuscript is sloppy and the article itself contains many errors and inaccuracies.
In the Abstract,
line 20 the authors write that "The Kriging interpolation method was used to interpolate the groundwater depth data of 106 monitoring wells in the Ulan Buh Desert over the past 20 years, and the spatial distribution characteristics of groundwater depth in the Ulan Buh Desert were analysed." In the next part of the article, there is no reference to research related to the fluctuation of the water table in these 106 wells. Please explain
line 27 and 28. Please provide a specific value in metres
Lines 31 – 34 this fragment is illegible, what did the authors mean?
Line 41, 42, 46, 47, 48, 51, 62, 70; 78 etc. such a citation is incorrect [2][3][4] correct version [2-4]
This shows the negligence of the authors.
Giving a reference to a website for meteorological data is a mockery of the reader.
It is impossible to refer to the vegetation cover trend without presenting specific rainfall data that could be provided in the supplement.
Line 292. Explain a burial depth, is it water or a human grave? Because when it comes to water, the language needs to be corrected.
Lines 293295, 296297 correct, the units is 1280.36m2 it should be 1280.36m2
Line 312. If we refer to the results presented in Figure 6 in Section 3.3, why is Figure 6 in another Section 3.4.1.?
Figure 7 in what units does vegetation cover?
Figure 8. I have never seen average temperatures in % in my life. Is this a new artificial intelligence???
Line 353, 354 units to be improved
The reception would be much better if the authors for selected places on the charts presented the diversity of vegetation cover in relation to the level of the groundwater table.
It is well known that the shallower the water, the better the vegetation cover, and there is nothing innovative about it.
Line 398 to 401 please explain "Except for 2010, the vegetation coverage in other years showed an upward trend compared to the previous period, and the growth rate of vegetation coverage was the fastest from 2000 to 2000, with a growth rate of 11. 88 % per decade.
In conclusion, the article by title is interesting but requires the authors to carefully and seriously improve the text and analyse the data. The conclusions are not consistent with the research results presented in the manuscript.
In its current form, the manuscript is suitable for a thorough improvement!!
The article "Effects of groundwater depth on vegetation coverage in Ulan Buh Desert in recent 20 years" presented for review is interesting; the collection of data from the 20th anniversary seems impressive.
However, the preparation of the manuscript is sloppy and the article itself contains many errors and inaccuracies.
In the abstract,
line 20 the authors write that "The Kriging interpolation method was used to interpolate the groundwater depth data of 106 monitoring wells in the Ulan Buh Desert over the past 20 years, and the spatial distribution characteristics of groundwater depth in the Ulan Buh Desert were analysed." In the next part of the article, there is no reference to research related to the fluctuation of the water table in these 106 wells. Please explain
line 27 and 28. Please provide a specific value in metres
Lines 31 – 34 this fragment is illegible, what did the authors mean?
Line 41, 42, 46, 47, 48, 51, 62, 70; 78 etc. such a citation is incorrect [2][3][4] correct version [2-4]
This shows the negligence of the authors.
Giving a reference to a website for meteorological data is a mockery of the reader.
It is impossible to refer to the vegetation cover trend without presenting specific rainfall data that could be provided in the supplement.
Line 292. Explain a burial depth, is it water or a human grave? Because when it comes to water, the language needs to be corrected.
Lines 293295, 296297 correct, the units is 1280.36m2 it should be 1280.36m2
Line 312. If we refer to the results presented in Figure 6 in Section 3.3, why is Figure 6 in another Section 3.4.1.?
Figure 7 in what units does vegetation cover?
Figure 8. I have never seen average temperatures in % in my life. Is this a new artificial intelligence???
Line 353, 354 units to be improved
The reception would be much better if the authors for selected places on the charts presented the diversity of vegetation cover in relation to the level of the groundwater table.
It is well known that the shallower the water, the better the vegetation cover, and there is nothing innovative about it.
Line 398 to 401 please explain "Except for 2010, the vegetation coverage in other years showed an upward trend compared to the previous period, and the growth rate of vegetation coverage was the fastest from 2000 to 2000, with a growth rate of 11. 88 % per decade.
In conclusion, the article by title is interesting but requires the authors to carefully and seriously improve the text and analyse the data. The conclusions are not consistent with the research results presented in the manuscript.
In its current form, the manuscript is suitable for a thorough improvement!!
Reviewer 2 Report
This manuscript uses a new tool to analyze vegetation coverage by correlating NDVI with groundwater depth in a desert area where almost all the vegetation is groundwater dependent. These studies are used in the literature from 10 years ago, but the innovation of this paper is the use of a new tool called Pixel Information Expert Engine. I recommend using the tool in at least a reduced area to calibrate the values of vegetation coverage because just by visual inspection using Google Earth, the vegetation coverage in the study area is far less than the classification shown in the paper.
General comments:
Improve the presentation of the software and include an example in order to give support to your conclusions.
Improve the graphs to give the information needed to analyze the changes in vegetation coverage and groundwater depth. I gave you some specific comments.
Rewrite the Results and Discussion section to provide elements for the conclusion section.
In conclusion number three, vegetation coverage and groundwater depth changes are caused mainly by human activity; precipitation and temperature have a moderated impact. Please analyze the extension of this impact in your study area and discuss it in the discussion section to provide the basis for the conclusion.
1. The manuscript needs Figure 1 with the location of the study area in a broad sense, which can be located in the Asian continent or at least in China. Indicate the climatological stations and monitoring well's location.
2. Figure 2, the vertical axis does not have units.
4. figure 7 does not have units for vegetation coverage. Could it be a percentage?
2. In Figure 8, please use two y-axes for the % of changes and the other for the data values. Extend the data to 2020 for comparison with vegetation coverage.
3. Figure 9. It must be drawn as bars because the value is annual. Remove the graph's title and include this information in the foot of the graph. Take care because precipitation units are (m), not (mm). This is not the trend of precipitation chances; the title could be: Average annual precipitation from 2000 to 2020—units in meters.
3. A map of groundwater depth evolution could be helpful in explaining the temporal variations.
Specific comments:
(Page 3, Line 106) Indicate the minimum and maximum temperature because the average doesn't give enough information about the extremes. See this information in: Niu, Y., Ren, G., Lin, G., Di Biase, L., & Fattorini, S. (2020). Fine-scale vegetation characteristics drive insect ensemble structures in a desert ecosystem: The tenebrionid beetles (Coleoptera: Tenebrionidae) inhabiting the Ulan Buh Desert (Inner Mongolia, China). Insects, 11(7), 410.
(Page 3, Line 107) are you referring to annual potential evaporation?
(Page 3, Line 120) Indicate in a figure the location of 107 monitoring wells. There is a discrepancy because on page 6, line 222 you indicate 106 wells.
(Page 6, Line 233) The explanation of formula six is missing the meaning of the subscript i. Put the subscript to R to get a correlation coefficient for every pixel, and explain the meaning of R.
(Page 6, Line 240) The value of 0.3 is not in Figure 2.
(Page 6, Line 243) The value of 0.3377 is not in Figure 2.
(Page 6, Line 246) Correct the period 2000 to 2000.
(Page 7, Line 263) Figure 4 is titled as absolute spatial distribution map of vegetation coverage, but in the legend of the map, the degree of spatial change in vegetation coverage and the classification is about it.
(Page 8, Line 284) There are only four time periods. It could be better: The data for each year selected were interpolated…
(Page 8, Lines 292-293) Improve the writing "area of areas" is confusing.
(Page 8, Line 293) indicate the increasing area by adding the percentage.
(Page 8, Lines 292-299) Do not use decimals in areas because of the resolution of pixels; they do not have meaning.
(Page 8-9, Lines 302-304) These ideas are not clear; please rewrite them.
(Page 9, Lines 315-318) This analysis is the most important part of this paper, extending the explanation of how the correlation was done, indicating that the changes in groundwater depth and vegetation coverage are for the entire period (2000-2020).
(page 9, Lines 319-321) Data to produce Figure 7 is from the measured wells, or how were these points selected?
(page 9, Lines 319-331) In order to discuss this scatter of points, it is necessarily identified the type of vegetation is covering. Various groundwater-dependent plants can reach water from depths of more than 20 m.
(Page 10, Line 334-335) The influence of temperature changes on vegetation coverage is analyzed and discussed in several papers; please refer to them to give a stronger argument.
(Page 11, Line 349-359) This information about the relationship between rain and groundwater depth must be presented before changes in groundwater depth are analyzed. On the other hand, I miss at least a brief explanation of what hydrological processes could produce groundwater fluctuation. I recommend including in Figure 9 the average groundwater depth for each year, which could help understand the hypothesis.
(Page 12, Line 390) The explanation of the acronym PIE in line 83 (page 2) was PIE (Pixel Information Expert Engine), and in this line (390), you indicate "Engine" additionally; please change to the same. In page 4, you use PIE platform. Use the same description for the same tool.
(Page 12, Lines 390-393) As I commented before, using this platform is a very important part of the paper, but you don't analyze the validity of the platform's results. It could be essential to exercise a small zone to identify different vegetation coverage in the field and compare it with the information the platform obtains. On the other hand, you do not explore the mechanism of groundwater depth to vegetation coverage you correlate changes in vegetation coverage with changes in groundwater depth.
(Page 12, Lines 394-401)
(Page 12, Line 397-401) Do not use 2 decimals to express areas that were estimated with a more significant error.
(Page 12, Line 401) Correct the period 2000 to 2000.
(Page 12, Line 401-402) The temperature trend is hard to explain because no data from 2017 to 2020 exists. If the temperature trend is conserved, then It can say that temperature is rising from 2000 to 2020 with some oscillations over the years, but the overall trend is increasing. Precipitation has been oscillating over the years, but the overall trend is maintaining 250 mm/year, with the most significant pick in 2007. On the other hand, the vegetation coverage presented a reduction area from 2005 to 2015.
(Page 12, Line 403-417) Repeat the same idea in lines 403-404 and 412-413. In lines 406-407, "The low vegetation coverage covers the high vegetation coverage" is confusing. Line 416 you comment about moderate variations that almost cover the entire Yellow River basin and salt lakes, but the Yellow River basin is entirely different in shape and size of your study area; it is hard to understand the relation.
(Page 12, Line 427-430) Relating precipitation and temperature to vegetation coverage and groundwater depth from the information analyzed is challenging. Additionally, the changes in land uses represent just a minimal area of the study area; It is not clear why this could be human activity reflected in the coverage of the study area. Maybe you can analyze the amount of water extracted and correlate it with the fluctuations of the groundwater levels; this could be a human impact on the vegetation coverage.
Reviewer 3 Report
-It is not clear how the VFC thresholds were defined, the low coverage range is very small compared to the ranges of the other levels
-The period analyzed is short to relate the changes in vegetation coverage with the depth of groundwater, there are only 4 years of information.
-The study doesn't have a validation analysis of the results
-It is clear that the vegetation coverage improves the recharge of groundwater, but it is not the only factor: the permeability, hydraulic conductivity, soil, among others, are not considered by the authors, nor do they present a conceptual model that allows to identify the directions of the water flow. underground in the study area.
-It is an error to make correlations at the spatial level by categorical regions (groundwater depth), they can lead to an inadequate interpretation of what happens in reality, especially when it is observed that the area >6m is much smaller in the study area, therefore it is unrepresentative, it is evidenced in the figure 6.
-The authors had to incorporate in a deeper way in the analysis, the precipitation and evapotranspiration.
-It is recommended that they evaluate the effect of vegetation coverage and groundwater depth using a hydrological model such as SWAT, CEQUEAU, INVEST-seasonal-wateryield, among others.
no comment